

# Methane Emissions from the Munich Oktoberfest

Jia Chen[1,*], Florian Dietrich[1,*], Hossein Maazallahi[2,4], Andreas Forstmaier[1], Dominik Winkler[1], Magdalena E. G. Hofmann[3], Hugo Denier van der Gon[4], and Thomas Röckmann[2]

[1]Environmental Sensing and Modeling, Technical University of Munich (TUM), Munich, Germany
[2]Institute for Marine and Atmospheric Research, Utrecht University, Utrecht, The Netherlands
[3]Picarro B.V., 's-Hertogenbosch, The Netherlands
[4]Climate, Air and Sustainability, TNO, Utrecht, The Netherlands
[*]Equal contribution
**Correspondence:** Jia Chen (jia.chen@tum.de), Florian Dietrich (flo.dietrich@tum.de)

**Abstract.**

This study presents the first investigation of the methane ($CH_4$) emissions of a big festival. In 2018 we measured the $CH_4$ emissions of Munich Oktoberfest, the world's largest folk festival, using in-situ measurements combined with a Gaussian plume dispersion model. Oktoberfest is a potential source for $CH_4$ as a high amount of natural gas for cooking and heating is

used.

Measurements were performed by walking and biking around the perimeter of the Oktoberfest premises (Theresienwiese) at different times of the day, during the week and at the weekend. The measurements show enhancements of up to 100 ppb compared to background values and measurements performed after Oktoberfest. The average emission flux of Oktoberfest is determined as $(6.7 \pm 0.6)\,\mu g/(m^2 s)$. Additional analyses, including the daily emission cycle and comparisons between

emissions and the number of visitors, suggest that $CH_4$ emissions of Oktoberfest are not only due to the human biogenic emissions; it is likely that fossil fuel $CH_4$ emissions, such as incomplete combustion or loss in the gas appliances, are the major contributors to Oktoberfest emissions.

Our results can help to develop $CH_4$ reduction policies and measures to reduce emissions at festivals and other major events in cities. Furthermore, events with a limited duration have not yet been included in the state-of-the-art emission inventories,

such as TNO-MACC, EDGAR or IER. Our investigations show that these emissions are not negligible. Therefore, these events should be included in future emission inventories.

## 1 Introduction

Climate change is a global problem that has a profound impact on living conditions and human societies. It is very likely that the present global warming is due to strong anthropogenic greenhouse gas (GHG) emissions. The Paris Agreement establishes

an international effort to limit the temperature increase to well below $2\,°C$ above pre-industrial levels. A "global stocktake" will revisit emission reduction goals every five years starting in 2023. The EU aims to cut its GHG emissions by $40\,\%$ by 2030 and by $80\,\%$ to $95\,\%$ by 2050, compared to the 1990 level. The German climate action plan (Klimaschutzplan 2050) contains similar goals, i.e. to cut at least $55\,\%$ of German GHG emissions by 2030 and at least $80\,\%$ by 2050.



Methane ($CH_4$) is the second-most prevalent GHG emitted by human activities (Allen et al. (2018); Etminan et al. (2016); Myhre et al. (2013)) and is estimated to have a global warming potential that is 28 to 34 times larger than that of $CO_2$ over the 100-year-horizon (IPCC (2013)). $CH_4$ has been responsible for around $20\%$ of the global warming by anthropogenic green-house gases since 1750 (Nisbet et al. (2014); Kirschke et al. (2013)). Current atmospheric $CH_4$ concentrations are 2.5 times as

high as the pre-industrial levels and its concentration growth is 3 times faster than $CO_2$ since the industrial revolution. After experiencing a nearly constant $CH_4$ concentration (total amount of $CH_4$ in the atmosphere) from 1999-2006, $CH_4$ concentrations have started to increase again (Saunois et al. (2016); Nisbet et al. (2014)). The reasons for the renewed growth are not fully understood; fossil fuel methane emissions are largely underestimated (Schwietzke et al. (2016)) and could play a major role in it (Hausmann et al. (2016); Worden et al. (2017)). Natural gas is a growing source of energy, but its unwanted release into the

atmosphere is a significant component of anthropogenic $CH_4$ emissions (Schwietzke et al. (2014); McKain et al. (2015)) and its reduction may be essential for attaining the goal of the Paris agreement.

Therefore, recent investigations have concentrated on detecting and quantifying $CH_4$ emissions from city gas pipelines, power plants as well as other gas and oil facilities using various methods. Phillips et al. (2013) mapped $CH_4$ leaks across all urban roads in the city of Boston using a cavity ring-down mobile analyzer. They identified 3,356 leaks with concentrations

exceeding up to 15 times the global background level, and used their isotopic signatures to show that the leaks are associated with natural gas. Roscioli et al. (2015) described a method using dual-tracer flux ratio measurements complemented by on-site observations to determine $CH_4$ emissions from natural gas gathering facilities and processing plants. Chen et al. (2017) and Toja-Silva et al. (2017) used differential column measurements (Chen et al. (2016)) and a computational fluid dynamics (CFD) model to quantify emissions from a natural-gas-based power plant in Munich. Atherton et al. (2017) conducted mobile

surveys of $CH_4$ emissions from oil and gas infrastructure in northeastern British Columbia, Canada and used the $CO_2/CH_4$ ratios to identify these emissions. Weller et al. (2018) evaluated the ability of mobile survey methodology (von Fischer et al. (2017)) to find natural gas leaks and quantified their emissions. Yacovitch et al. (2015) measured $CH_4$ and ethane ($C_2H_6$) concentrations downwind of natural gas facilities in the Barnett shale region using a mobile laboratory. A couple of years later, Yacovitch et al. (2018) investigated the Groningen natural gas field, one of Europe's major gas fields using their mobile

laboratory in combination with airborne measurements. Luther et al. (2019) deployed a mobile sun-viewing Fourier transform spectrometer to quantify $CH_4$ emissions from hard coal mines. Other studies laid a special focus on city and regional emissions of fossil fuel $CH_4$. McKain et al. (2015) determined natural gas emission rates for the Boston urban area using a network of in-situ measurements of $CH_4$ and $C_2H_6$ and a high resolution modeling framework. Lamb et al. (2016) quantified the total $CH_4$ emissions from Indianapolis using the aircraft mass balance method and inverse modeling of tower observations, and

distinguished its fossil fuel component using $C_2H_6/CH_4$ tower data. Wunch et al. (2016) used total column measurements of $CH_4$ and $C_2H_6$ recorded since the late 1980s to quantify the loss of natural gas within California's South Coast Air Basin. Most recently, Plant et al. (2019) reported aircraft observations of $CH_4$, $CO_2$, $C_2H_6$, and carbon monoxide (CO) of six old and leak-prone major cities along the East Coast of the United States. They found emissions attributed to natural gas are about a factor of 10 larger than the values provided by the EPA inventory.





Large folk festivals are also likely sources of anthropogenic emissions of air pollutants, such as nitrogen oxides ($NO_x$), CO, particulate matter (PM2.5, PM10), Sulfur dioxide ($SO_2$), etc. Huang et al. (2012) investigated the impact of human activity on air quality before, during, and after the Chinese Spring Festival 2009, the most important festival in China. They used potential source contribution function analysis to illustrate the possible source for air pollutants in Shanghai. Shi et al. (2014)

measured concentrations of particulate matters and polycyclic aromatic hydrocarbons (PAHs) during the Chinese New Year's Festival 2013 and estimated the source attributions from cooking, vehicle, biomass and coal combustion. Kuo et al. (2006) investigated PAH and lead emissions from cooking during the Chinese mid-autumn festival. Nishanth et al. (2012) reported elevated concentrations of various air pollutants such as ozone ($O_3$), $NO_x$, and PM10 after the traditional Vishu festival in South India. Nevertheless, up to now, festivals have not been considered a significant $CH_4$ emission source and accordingly,

$CH_4$ emissions of festivals have not been studied.

Oktoberfest, the world's largest folk festival with over 6 million visitors annually, is held in Munich. In 2018, during the 16 days of Oktoberfest, approximately 8-million liters of beer were consumed. For cleaning, dish washing, toilet flushing, etc., about 100-million liters of water were needed. The use of energy added up to 2.9 million $kWh$ of electricity and $200{,}937\,m^3$ of natural gas, $79\,\%$ of which is used for cooking and $21\,\%$ for heating (München (2018a)).

The measurements during our 2017 Munich city campaign indicated Oktoberfest as a possible source for $CH_4$ for the first time (Chen et al. (2018)). For a better source attribution and a quantitative emission assessment, we have investigated the $CH_4$ emissions from Oktoberfest 2018 by performing mobile in-situ measurements and incorporating a Gaussian plume dispersion model. These measurements and modeling approaches are described in section 2. The results of these investigations show that Oktoberfest is an anthropogenic source of $CH_4$ that has not been accounted for until now. We have compared the

determined total emission flux with bottom-up estimates of biogenic emissions from human, and also present the daily cycle of the emissions. In addition, the week and weekend variations are shown. From these findings we can draw conclusions about the origins of the Oktoberfest $CH_4$ emissions, which are presented in section 3.

## 2  Method

We conducted a mobile survey around the perimeter of Oktoberfest to obtain the $CH_4$ concentration values around the festival

area (Theresienwiese) and incorporated a Gaussian plume model consisting of 16 different point sources to determine the $CH_4$ emission strength of Oktoberfest.

### 2.1  Measurement approach and instrumentation

The measurements include both $CH_4$ and wind measurements. The sensors and the way they are used are described in the following sections.



### 2.1.1 Concentration measurements

Mobile in-situ measurements were conducted to quantify $CH_4$ enhancements. To this end, two portable Picarro GasScouters G4302 for measuring $CH_4$ and $C_2H_6$ were used. The sensor is based on the cavity ring-down measurement principle (O'Keefe and Deacon (1988)), using a laser as a light source and a high-finesse optical cavity for measuring gas concentrations with

high precision, which for $CH_4$ detection is $3\,\mathrm{ppb}$ ($CH_4$ only mode) with $1\,\mathrm{s}$ integration time (Picarro (2017)). We applied a moving-average filter with a window size of $10\,\mathrm{s}$ and a step size of $5\,\mathrm{s}$ to the $1\,\mathrm{s}$ raw measurements. Since the data is averaged over $10\,\mathrm{s}$, the precision is improved to $1\,\mathrm{ppb}$. To distinguish between fossil-fuel related and biogenic emissions, the instrument can be switched to $CH_4/C_2H_6$ mode and measure $C_2H_6$ with a precision of $10\,\mathrm{ppb}$ for an integration time of $1\,\mathrm{s}$.

Since we were not allowed to enter the festival area due to safety concerns, the measurements were performed by walking

and biking around the perimeter of Oktoberfest right next to the security fences many times wearing the analyzer as a backpack. The measurements were taken on several days during and after the time of the festival to compare the differences in emission strength and distribution. To also observe the hourly dependency of the emissions, the measurements were distributed over the course of the day. In the end, we covered the period between 8:00 a.m. and 7:00 p.m. (local time) hourly.

For the study two identical GasScouters G4302 were deployed. One instrument was provided by TNO and the other one

from Picarro Inc. The former one was used in the first week while the latter one was used in the second week of Oktoberfest and for the time after the festival. Although the measurement approach is based on determining the enhancements and not on comparing absolute concentration values, calibration measurements between the two instruments were conducted during the campaign.

### 2.1.2 Wind measurements

In addition to the gas concentrations, wind measurements are vital for estimating the emission numbers of the Oktoberfest using atmospheric models. To this end, a 2D ultrasonic wind sensor (Gill WindObserver II) was located on a roof close by (48.148° N, 11.573° E, 24 m agl.). These wind measurements were utilized for the emission estimates.

To assess the uncertainty of the wind measurements, we compared these measurements with the values reported by an official station of Germany's National Meteorological Service (Deutscher Wetterdienst, DWD). The DWD station (48.163° N, 11.543°

E, 28.5 m agl.) is located about $2.8\,\mathrm{km}$ away. As this distance is about the radius of the Munich inner city, we assumed that the difference between the two stations is representative for the uncertainty of two arbitrary measurement points in the downtown area, which is also home to Oktoberfest.

### 2.2 Modeling approach

To quantify the emissions of Oktoberfest, we used the measured concentration values as input for an atmospheric transport

model.





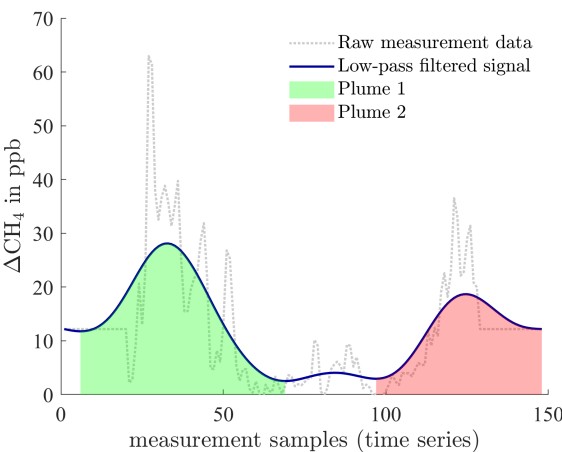

**Figure 1.** The pre-processed measurement signal (dotted line, moving average with window size $10\,\mathrm{s}$ and step size $5\,\mathrm{s}$) is shown along with a low pass filtered version (blue line), which is used to obtain the single plumes (green and red area). The signal in the center is not detected as a plume, as the enhancement is not high enough. The shown round was recorded by bike and took $750\,\mathrm{s}$ ($12.5\,\mathrm{minutes}$).

### 2.2.1 Selection Algorithm

For our modeling approach, the plumes of individual surveys (hereafter referred to as "rounds") around the Theresienwiese were evaluated. In total, we completed 94 rounds (69 during and 25 after Oktoberfest). For every round the individual plumes were determined by analyzing a low-pass filtered version of the measurement time series. A Kaiser window was utilized for the low-pass filtering.

Once the signal was filtered, a signal section between two adjacent valleys was defined as a plume signal if it had an enhancement of more than $5\,\mathrm{ppb}$. This process is illustrated in Figure 1.

When the initial plume selection phase was completed, the identified plumes were analyzed further. As the path of a measurement around the Oktoberfest premises was predefined by the security fence, the location of each point on that route can be converted into a fixed angle. This simplifies the comparison between the measurements and the model. For that reason, a center point of the Theresienwiese was defined (cf. green dot in Figure 2, $48.1315°$ N, $11.5496°$ E). With the help of this point, an angle was assigned to all measurement and model values. This angle was defined similarly to the wind angles, meaning that $0°$ is in the north and $90°$ is in the east.

In order to decide whether a measured plume is attributable to emissions from Oktoberfest, a forward model uses the measured wind direction (with uncertainty) to calculate at which angles a plume from Oktoberfest should occur. As can be seen in Figure 3, only plume 1 was selected because the angle range of this plume (green) overlaps mostly with the accepted angle range (grey) computed by the forward model of this plume. In contrast, plume 2 (red) has no overlap with the range computed by the forward model, hence plume 2 was discarded.





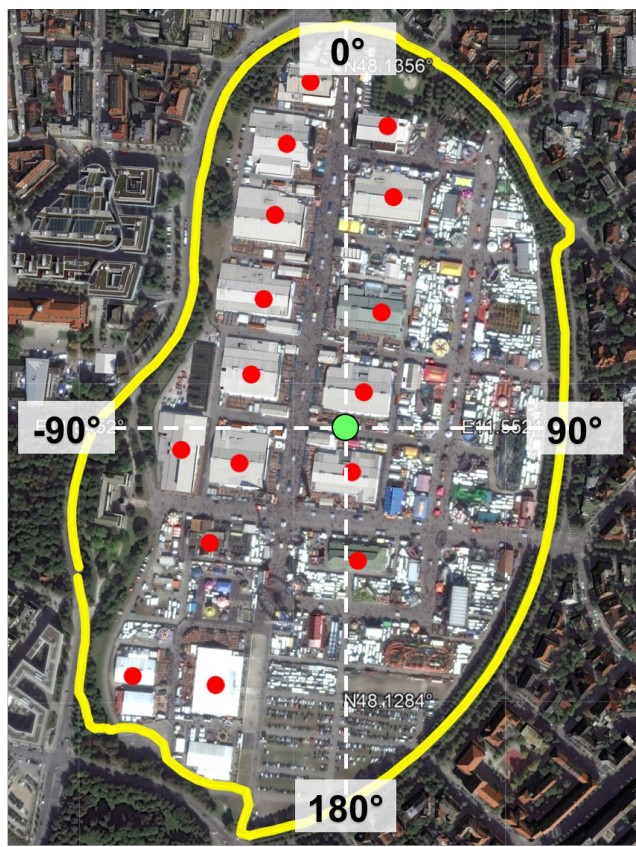

**Figure 2.** Standard route around Oktoberfest (yellow) including the locations of the 16 big tents (red) and the center point (green). Map data: © Google, DigitalGlobe

Additionally, the standard deviation of the wind direction over the time the plume was recorded is taken into account. If the standard deviation is higher than 25°, the plume is not considered, as our approach requires stable wind conditions.

The selection algorithm is visually summarized in Figure 4:





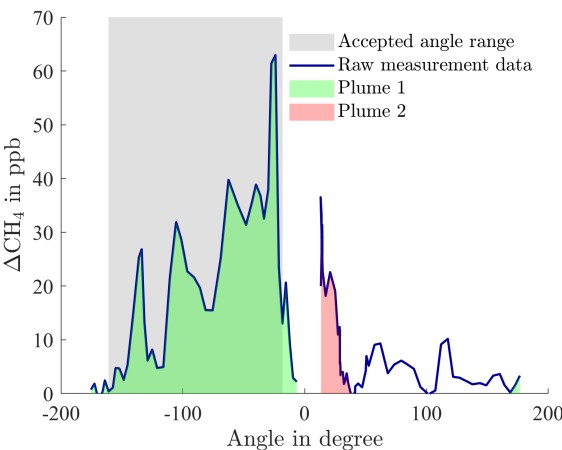

**Figure 3.** Measurement signal mapped onto the standard route with the angle on the abscissa. Two detected plumes and the accepted angle range computed by the forward model are highlighted. Plume 2 has no overlap with the accepted range and is therefore discarded.

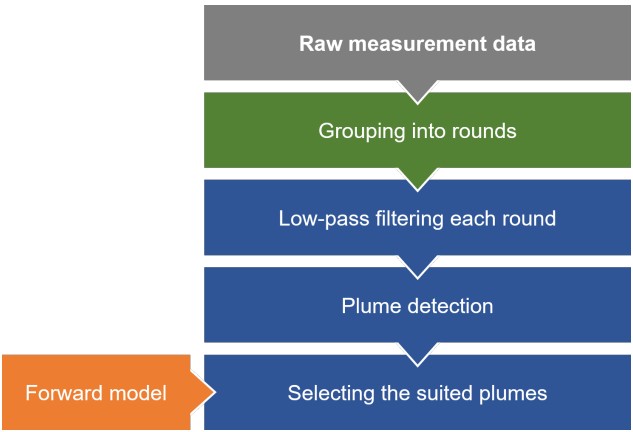

**Figure 4.** Flowchart visualizing the main steps performed on the raw measurement data in order to obtain an emission estimate.

### 2.2.2 Baseline determination

As one measurement round can take up to one hour (in case of walking), the atmospheric conditions can vary during that time period. This will result in a changing background concentration. Therefore, the baseline for determining the concentration enhancements cannot be calculated solely using a constant value.

5    In our approach, we assume that the baseline during one round is either rising or falling and that there is a linear behaviour. Such a straight line is clearly defined by two points. For that reason, the time series for each round was divided into two equally sized bins (first and second half). For each half, we determined the lowest $10\,\%$ quantile. Afterwards, the mean values of the $10\,\%$ smallest concentration values of each bin were used to define one straight line, which was used as the background for



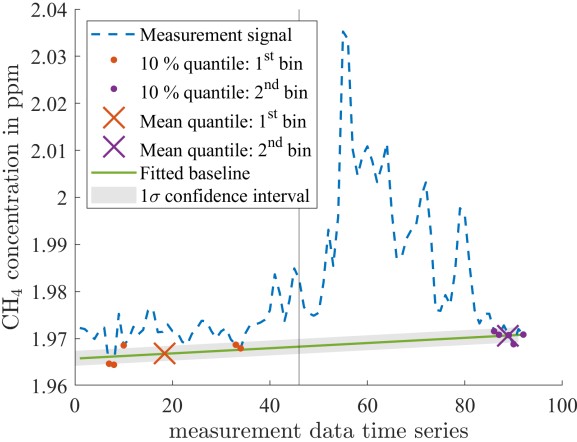

**Figure 5.** Baseline determination by dividing the measurement signal (blue) into two halves. Afterwards, a line (green) is fitted through the mean values of the $10\,\%$ lowest concentration points of each half. The grey shaded area denotes the $1\,\sigma$ uncertainty range of the background line.

that specific round (cf. Figure 5). The uncertainty of that baseline was determined by the $CH_4$ concentration deviations of the $10\,\%$ smallest values from the baseline.

### 2.2.3 Gaussian plume model

The framework of our modelling approach is based on a Gaussian plume model, which is described in Pasquill (1966, 1969,
5  1979); Gifford (1976); Briggs (1973); Hanna et al. (1982), and widely used in studies for assessing local source emissions (Bovensmann et al. (2010); Yacovitch et al. (2015); Atherton et al. (2017); Nassar et al. (2017); Kiemle et al. (2017)). It is a steady-state model that simulates the processes of diffusion and the transport of emitted trace gases from a point source. The gas disperses such that its concentration distributions fit well to Gaussian curves in vertical and horizontal directions.

For a point source emitting continuously with strength Q (unit: $\mathrm{mol\,s}^{-1}$) at effective height $H$ above the ground and uniform
10 wind speed, the expression for the time-averaged concentration $< c(x,y,z) >$ (unit: $\mathrm{mol\,m}^{-3}$) is given by the formula below:

$$< c(x,y,z) >= \frac{Q}{2\,\pi\,\overline{u}\,\sigma_y(x)\,\sigma_z(x)}\,\exp\left(-\frac{y^2}{2\,\sigma_y(x)^2}\right)\cdot\left(\exp\left(-\frac{(z-H)^2}{2\,\sigma_z(x)^2}\right)+\exp\left(-\frac{(z+H)^2}{2\,\sigma_z(x)^2}\right)\right) \tag{1}$$

with $x$, $y$ and $z$ describing the downwind distance, horizontal/cross-wind distance to the $x$ axis and the height above the ground, respectively. $\overline{u}$ is the time-averaged wind speed, $\sigma_y(x)$ is the standard deviation of the concentration in the cross-wind direction and $\sigma_z(x)$ is the standard deviation of the concentration in the vertical direction. These dispersion coefficients
15 describe the spreading of the plume increasing with the downwind distance from the source $x$.

To determine the dependency of $\sigma_y$ and $\sigma_z$ on $x$, diffusion experiments were carried out (Haugen et al. 1958), which resulted in Pasquill's curves (Pasquill (1979); Gifford (1976)). Smith (1968) worked out an analytic power-law formula for





the relationship between $\sigma_y$, $\sigma_z$ and $x$. Briggs (1973) combined the aforementioned curves and used theoretical concepts to produce the widely used formulas given in Hanna et al. (1982).

During the measurement periods, the surface wind was lower than $4\,\mathrm{m\,s^{-1}}$ and the insolation was strong to moderate. Therefore, stability class A or B was chosen according to the Pasquill turbulence types (Gifford (1976)).

Based on the recommendations by Briggs for urban conditions (Briggs (1973); Hanna et al. (1982)), the relationships between the dispersion parameters and the downwind distance are described as:

$$\sigma_y(x) = 0.32\,x\,(1 + 0.0004\,x)^{-1/2}, \tag{2}$$

$$\sigma_z(x) = 0.24\,x\,(1 + 0.001\,x)^{1/2}. \tag{3}$$

Those relationships were used in our study.

### 2.2.4   Multiple Gaussian plume model

The concentration measurements using the backpack instrument were performed close to the festival area ($< 500\,\mathrm{m}$), which is why the emissions of Oktoberfest cannot be seen as a single point source. For this reason multiple point sources were used. All these point sources were modelled using Gaussian plumes before they were superimposed. The spatially superimposed plumes

were detected as a continuous plume signal in our measurement. Later on, these plume signals were utilized for the emission assessment.

Since the emission sources of Oktoberfest were unknown, the locations with the highest density of visitors and with the highest energy consumption were chosen as main sources for the model. Those locations are represented by the 16 biggest beer tents ($> 1{,}000$ seats) on the festival premises (cf. red dots in Figure 2). To achieve a good correlation between the model

and reality, these 16 tents were not treated equally in the final model. Instead, they were linearly weighted according to their maximum number of visitors. Therefore, the biggest tent (about 8,500 visitors) has, in the end, a more than eight times higher influence on the total emission number than the smallest one (about 1,000 visitors).

### 2.2.5   Forward modelling approach

The aforementioned multiple Gaussian plume model was used in a forward approach to compare the measured and modeled

concentration signals with each other. For that, a predefined route around Oktoberfest was used (cf. yellow route in Figure 2) to determine the concentrations for each angle.

The actual shape of the concentration vs. angle graph $c(\alpha)$ for every selected plume $i$ is considered for the determination of the emission number (cf. Figure 6, blue curve). The optimization procedure can be expressed mathematically as follows:

$$E_{\mathrm{Okt},i} = \arg\min_{E_i} \int_0^{360} |c(\alpha) - M(E_i, \alpha)|\,\mathrm{d}\alpha, \tag{4}$$





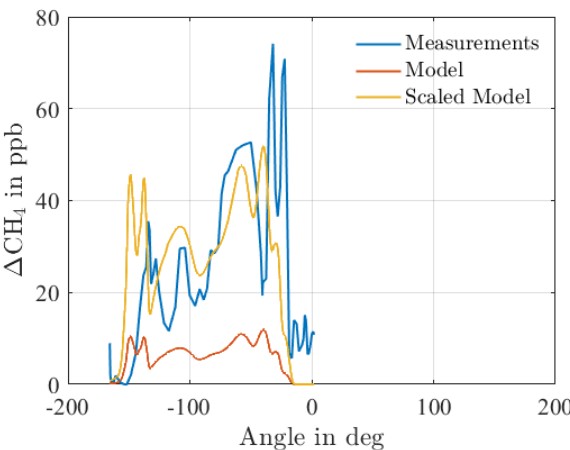

**Figure 6.** Raw measurement curve (blue) with the a priori forward model (orange) and the scaled forward model (yellow).

where $M$ represents the model. The emission number of Oktoberfest $E_i$ was varied until the areas underneath the modelled and measured curves are the same, and thus the sum of the absolute difference between the model and measurement is minimized.

Practically, we computed the forward model using the averaged wind information at this time and a prior emission number $E_{prior}$ of $1\,\mathrm{g\,s^{-1}}$ and compared it with the measurement curve. In case the shape looks similar (high cross-correlation coeffi-
5    cient), a scaling factor is applied to the prior emission number and varied until the forward model matches the measurements. This procedure is illustrated for one exemplary plume signal in Figure 6. There, the prior modelled concentrations (orange) are smaller than the measured concentrations (blue). Therefore, the model has to be multiplied with a scaling factor until the areas underneath the modelled and measured curve are the same (yellow). By multiplying the scaling factor $k_{scaling,i}$ with the $E_{prior}$, the emission number of Oktoberfest $E_{Oktoberfest,i}$ for every plume signal $i$ can be determined as:

$$E_{Okt,i} = E_{prior} \cdot k_{scaling,i}. \tag{5}$$

### 2.2.6   Uncertainty assessment

To determine the uncertainty of the final emission numbers, we considered the uncertainties of our input parameters. These include uncertainties in the wind and concentration measurements as well as uncertainties in the determined baseline. These input parameters were modelled as Gaussian distributions each. Afterwards, the emission number was determined by running
15   our modeling approach 1,000 times using those four parameters (wind speed, wind direction, measured $CH_4$ concentration, background concentration) as input. In each round, slightly different input values were chosen randomly and independent from each other out of those four distributions.

The concentration measurement uncertainty is indicated by the manufacturer Picarro to about $1\,\mathrm{ppb}$ for an averaging time of $10\,\mathrm{s}$. This value was used as the standard deviation of the modelled input distribution.





**Table 1.** Mean and standard deviation of the input parameters for the $CH_4$ plume signal $i$

| Type | Mean | Standard deviation |
|---|---|---|
| Wind speed | $v_{\text{wind,meas},i}$ | $0.5\,\text{m}\,\text{s}^{-1}$ |
| Wind direction | $\alpha_{\text{wind,meas},i}$ | $24°$ |
| Instrumentation | $c_{\text{meas},i}$ | $1\,\text{ppb}$ |
| Background | $c_{\text{backgnd},i}(t)$ | $\sigma_{10\%\,\text{quantile},i}$ |

For the wind speed and direction, not only the instrument uncertainty but also the spatial variations of the winds were taken into account. For that reason, the uncertainty of the wind measurements was determined by comparing two surface measurement stations within the inner city of Munich (cf. section 2.1.2). We determined the differences in wind speed and direction throughout September and October 2018. The differences are representative of the heterogeneity of the wind within

the inner city of Munich and, therefore, represent an upper bound for the uncertainty of the wind within the Oktoberfest premises. The comparison of both the wind speed and direction resulted in Gaussian shaped distributions with mean values each around zero. The standard deviations of the differences between the reported wind directions and speeds of the two stations are $24°$ and $0.5\,\text{m}\,\text{s}^{-1}$ throughout September and October 2018.

The baseline approach described in section 2.2.2 introduces a further error which has to be considered as well. The back-

ground concentrations were modelled as a Gaussian distribution where its standard deviation was calculated from the $CH_4$ concentration deviations between the 10% smallest values of each bin and the baseline shown in Figure 5.

The used parameters for the uncertainty assessment are summarized in Table 1.

## 3   Results and discussion

### 3.1   Concentration mapping

To show that there is a clear correlation between the wind directions and the enhancements, the measured $CH_4$ concentrations were plotted for each round on a map of the Oktoberfest premises. As the variations of the boundary layer height should not be taken into account, those plots are not showing the absolute concentration values but just the enhancements above the determined background concentration (cf. section 2.2.2).

Two examples of such plots for two different wind directions are shown in Figure 7. In addition to the concentration en-

hancements and the wind direction, the 16 emission sources are shown as black dots on top of each tent. The Gaussian plumes themselves are also represented qualitatively. Those two plots exemplify that the highest concentration enhancements can be observed downwind of the Oktoberfest premises.





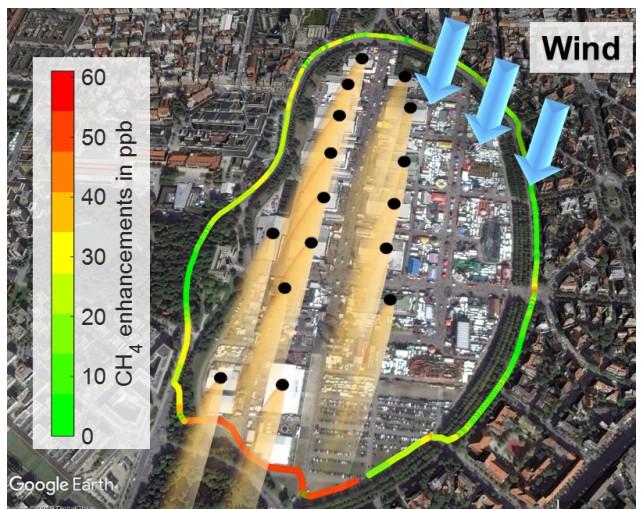

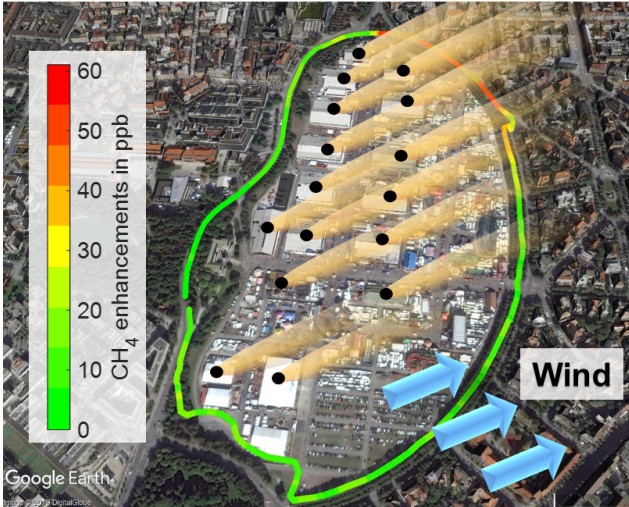

**Figure 7.** $CH_4$ concentration of one measurement round including the influence of the 16 Gaussian plumes of the tents (black dots).
Wind direction: upper panel 20°, lower panel −110°. Map data: © Google, DigitalGlobe





## 3.2 Emission number

The average emission number of the Oktoberfest 2018 $E_{\text{Okt, avg}}$ is determined by averaging the emission numbers of the $N$ plume signals $E_{\text{Okt,i}}$ accordingly:

$$E_{\text{Okt, avg}} = \frac{1}{N} \sum_{i=1}^{N} E_{\text{Okt,i}}. \tag{6}$$

To make the final emission number more robust and to be able to determine an uncertainty, the basic approach of Eq. 6 was improved. Instead of just using the actual measured data, an uncertainty range was applied to the four main input parameters using Gaussian distributions each (cf. section 2.2.6).

For every plume signal $i$, 1,000 samples of randomly chosen input datasets from the aforementioned normal distributions of the input parameters were used to determine 1,000 slightly different emission numbers $E_{\text{Okt, i, k}}$. Using Eq. 6, an average
emission number for each realization $E_{\text{Okt, avg, k}}$ was determined:

$$E_{\text{Okt, avg, k}} = \frac{1}{N} \sum_{i=1}^{N} E_{\text{Okt,i, k}}. \tag{7}$$

The average emission number including an uncertainty assessment can be obtained by determining the mean $\mu_{\text{Okt}}$ and standard deviation $\sigma_{\text{Okt}}$ of those 1,000 realizations:

$$\mu_{\text{Okt}} = \frac{1}{1000} \sum_{k=1}^{1000} E_{\text{Okt,avg, k}}, \tag{8}$$

$$\sigma_{\text{Okt}} = \sqrt{\frac{\sum_{k=1}^{1000} (E_{\text{Okt,avg, k}} - \mu_{\text{Okt}})^2}{999}}. \tag{9}$$

The result for the total emission number of Oktoberfest 2018 is shown in Figure 8 and has a value of

$$E_{\text{Okt,total}} = \mu_{\text{Okt}} \pm \sigma_{\text{Okt}} = (6.7 \pm 0.6)\,\mu\text{g/(m}^2\text{s)}. \tag{10}$$

To verify whether those emissions were caused by the Oktoberfest, Figure 8 also shows the determined emissions for the time
after Oktoberfest (from October $8^{th}$ through October $25^{th}$). This number $((1.1 \pm 0.3)\,\mu\text{g/(m}^2\text{s}))$ is significantly smaller than the one during Oktoberfest but still not zero. It indicates that the emissions are caused by Oktoberfest, and the disassembling of all the facilities, which takes several weeks, still produces emissions after Oktoberfest.

After grouping the emission numbers into the two categories weekday and weekend, two separated distributions are visible in Figure 9. The average emission for the weekend $((8.5 \pm 0.7)\,\mu\text{g/(m}^2\text{s}))$ is higher than the averaged emission for the weekday
$((4.6 \pm 0.9)\,\mu\text{g/(m}^2\text{s}))$, almost by a factor of two. To interpret this behavior, the visitor trend of the Oktoberfest was investigated. This trend is based on the officially expected amounts of visitors (muenchen.de (2018)) and was linearly interpolated in between (see Figure 10). Besides the daily trend, it also shows the mean values of the week- and weekend days (dotted lines). As the number of visitors at Oktoberfest was significantly higher on a weekend day ($\approx 1.4$) than on a weekday ($\approx 0.75$) (cf. Figure 10), it shows that a higher number of visitors results in higher emission, which indicates the CH$_4$ emissions are
anthropogenic.





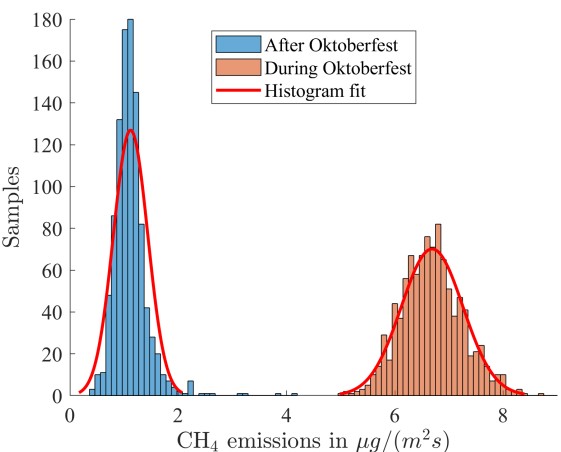

**Figure 8.** Total CH$_4$ emission estimate during (light red) and after (blue) the Oktoberfest 2018 including a fitted normal distribution (red line).

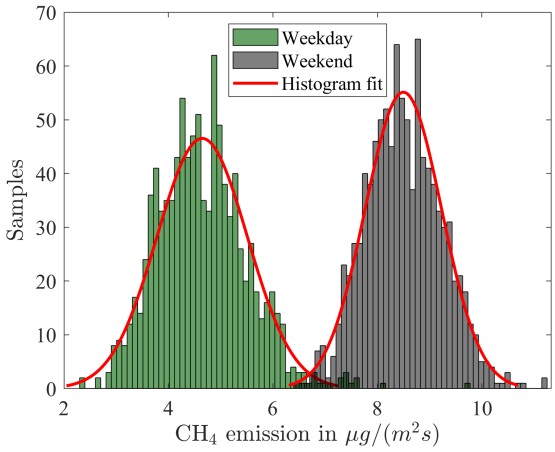

**Figure 9.** CH$_4$ emission estimates for a weekday (green) and a weekend day (black) including a fitted normal distribution (red line).





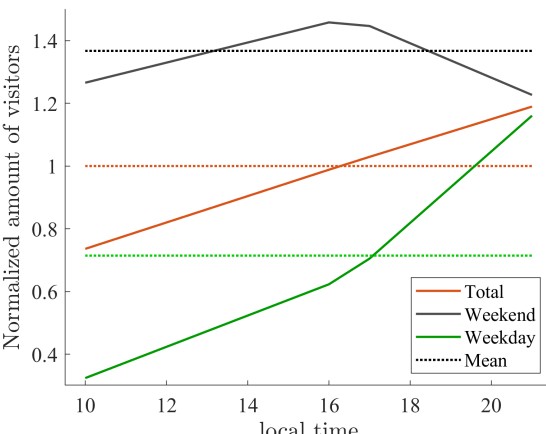

**Figure 10.** Qualitative daily trend of the number of visitors at Oktoberfest distinguished for the weekend (black), weekday (green), and total (red). The dotted line represents the mean value of each trend line.

### 3.3 Daily emission cycle

To assess the daily cycle of the $CH_4$ emissions, the emission numbers of the plume signals $E_{Okt,i,k}$ are grouped into hourly bins. Then for each bin an average emission number $E_{Okt,hour,k}$ is calculated. Afterwards, these numbers are averaged for the 1,000 realizations to get robust numbers including an uncertainty estimate:

$$\mu_{\text{Okt, hour}} = \frac{1}{1000}\sum_{k=1}^{1000} E_{\text{Okt, hour},k}, \tag{11}$$

$$\sigma_{\text{Okt,hour}} = \sqrt{\frac{\sum_{k=1}^{1000}(E_{\text{Okt,hour,k}} - \mu_{\text{Okt,hour}}.)^2}{999}} \tag{12}$$

In Figure 11, the variation of the hourly emission mean ($\mu_{\text{Okt, hour}}$) is shown as a blue line. The grey shaded area shows the uncertainty ($\sigma_{\text{Okt,hour}}$) of the emission numbers within that hour. The daily emission cycle shows an increasing trend towards the evening superimposed with an oscillating behavior.

The linear increasing trend agrees with Figure 10, which shows a linearly increasing visitor amount throughout the day, affirming the anthropogenic nature of the emission. The oscillating behavior indicates that the emissions are related to time-dependent events such as cooking, heating and cleaning, which tends to have the maxima in the morning, noon and evening.

### 3.4 Biogenic human $CH_4$ emissions

To address the question whether the people themselves caused the emissions or whether the emissions were caused by processes that are related to the number of visitors, such as cooking, heating, sewage, etc., we took a closer look at human biogenic emissions.




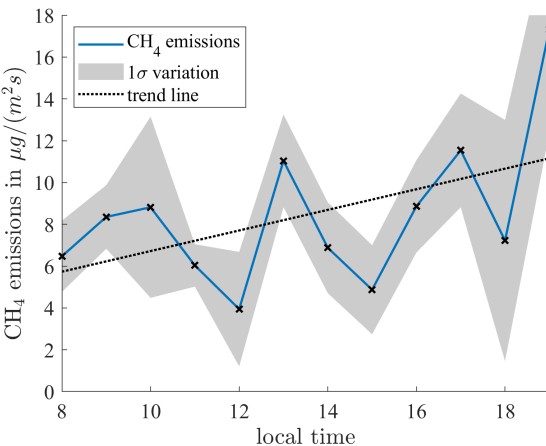

**Figure 11.** Daily variations of the emissions from the Oktoberfest. The grey shaded area denotes the variation (1 $\sigma$ standard deviation) within that hour.

According to de Lacy Costello et al. (2013), $CH_4$ is produced in amounts measurable in the breath of up to $50\,\%$ of the population. Keppler et al. (2016), however, used laser absorption spectroscopy to confirm that all humans exhale $CH_4$. The mean of the breath $CH_4$ enhancements above the background from all test persons in Keppler et al. (2016) (112 persons with an age range from 1 to 80 years) is $2316\,\mathrm{ppb}$. We multiplied this value with the 300,000 persons that visit the Oktoberfest premises ($\approx 3.45 \cdot 10^5\,\mathrm{m^2}$) every day. This represents an upper limit of people who are at the Theresienwiese at the same time, as most visitors do not stay all day long. A typical human respiratory minute volume is $6\,\mathrm{L/min}$ (Gedeon (2009)). The gas density of $CH_4$ at $20\,°C$ and $1\,\mathrm{atm}$ is $0.67\,\mathrm{g\,L^{-1}}$. Therefore, the expected $CH_4$ emission from the human breath can be calculated as:

$$E_{\text{breath}} = \frac{2316\,\mathrm{ppb} \cdot 300000 \cdot \frac{6\,\mathrm{L}}{60\,\mathrm{s}} \cdot 0.67\,\frac{\mathrm{g}}{\mathrm{L}}}{3.45 \cdot 10^5\,\mathrm{m^2}} = 0.13\,\frac{\mu\mathrm{g}}{\mathrm{m^2\,s}}. \tag{13}$$

Considering that about $80\,\%$ of $CH_4$ is excreted by flatulence and about $20\,\%$ in one's breath (de Lacy Costello et al. (2013)), the $CH_4$ emission from the human breath and flatulence is in total around

$$E_{\text{human biogenic,total}} = \frac{0.13\,\frac{\mu\mathrm{g}}{\mathrm{m^2\,s}}}{20\,\%} = 0.65\,\frac{\mu\mathrm{g}}{\mathrm{m^2\,s}}. \tag{14}$$

Although, we assumed the maximum possible number of visitors, the calculated biogenic emission number is more than one magnitude smaller than the emission number we determined for Oktoberfest in this study. Therefore, the determined emissions are not directly produced by the humans themselves, but by processes that are related to the visitor amount.

### 3.5  Emission from sewage

Besides the direct biogenic human emissions, $CH_4$ emissions out of sewer systems are possible sources. Those emissions are a product of bacterial metabolism within the waste water. Its emission strength depends especially on the hydraulic retention time





(Liu et al. (2015); Guisasola et al. (2008)) which represents the time the waste water stays in the system. This time decreases with a higher amount of waste water, as the flow increases in such a case.

At Oktoberfest, the amount of waste water is very high as the 100-million liters of consumed water and the 8-million liters of beer have to flow into the sewer system at some time (München (2018b)). Therefore, the retention time in the sewer system

underneath the Theresienwiese is quite low, which makes high $CH_4$ emissions from sewage unlikely. Furthermore, the waste water consists mainly of dirty water and urine, which does not contain a lot of carbon compounds that are necessary to produce $CH_4$.

### 3.6   Fossil fuel $CH_4$ emissions

The biogenic emissions can likely not fully explain the determined emission number of Oktoberfest. Therefore, fossil-fuel

related emissions have to be considered as well. According to the weekday/weekend emission comparison (cf. Figure 9) and the daily emission cycle (cf. Figure 11 compared with Figure 10), there is, in general, a visitor-dependent linear increase of $CH_4$ emission throughout the day that is superimposed with time-dependent events such as cooking, cleaning or heating. These events can cause $CH_4$ emissions, as about $40\%$ of the used energy at Oktoberfest is provided by natural gas that is used for cooking ($79\%$) and heating ($21\%$).

As the human biogenic $CH_4$ emissions have already been excluded due to too small values, leakages and incomplete burning in the gas appliances provide a possibility to explain the emissions. Ethane is a tracer of thermogenic $CH_4$, and can be used to indicate a natural gas related source (Yacovitch et al. (2014); McKain et al. (2015)). For that reason, we deployed a portable instrument that is designed to measure $CH_4$ but that is also capable of measuring $C_2H_6$. Due to the aforementioned safety reasons, the distance between the measurements and the closest point source (tent) was $50\,\mathrm{m}$ to $250\,\mathrm{m}$. Therefore, the $CH_4$

concentration was relatively low (max. $100\,\mathrm{ppb}$). According to the Munich municipal utilities, the $C_2H_6/CH_4$ ratio of natural gas used in Munich is about $3\%$ (München (2018c)). This results in an $C_2H_6$ concentration lower than 3 ppb, assuming that all of the measured $CH_4$ is sourced from natural gas. Such a small concentration value is lower than the detection limit of the GasScouter (about $3\,\mathrm{ppb}$ with $10\,\mathrm{s}$ integration time), which is why we were not able to determine the $C_2H_6/CH_4$ ratio of the measured gas.

Nevertheless, it is possible to determine an upper bound for the loss rate of natural gas if one assumes that all the emissions are fossil-fuel based.

The gas consumption at Oktoberfest 2018 sums up to $200{,}937\,\mathrm{m}^3$. Therefore, the total weight of the consumed $CH_4$ at Oktoberfest yields

$$M_{\text{gas,total}} = 0.668\,\mathrm{kg\,m^{-3}} \cdot 200{,}937\,\mathrm{m}^3 = 1.34 \cdot 10^5\,\mathrm{kg}. \tag{15}$$

In this study, the $CH_4$ flux of Oktoberfest has been determined to $6.7\,\mathrm{\mu g/(m^2 s)}$. If we assume that the emission is continuous throughout the day (about $11\,\mathrm{h}$ opening time per day) and homogeneous throughout the entire Oktoberfest premises, the total amount of $CH_4$ lost to the atmosphere would be:

$$M_{\text{CH4,loss,max}} = 6.7\,\frac{\mathrm{\mu g}}{\mathrm{m^2 s}} \cdot \left(16\,\mathrm{d} \cdot 11\,\frac{\mathrm{h}}{\mathrm{d}} \cdot 3600\,\frac{\mathrm{s}}{\mathrm{h}}\right) \cdot 3.45 \cdot 10^5\,\mathrm{m}^2 = 1.46 \cdot 10^3\,\mathrm{kg}. \tag{16}$$




**Table 2.** Comparison of the Oktoberfest emission flux to state-of-the-art emission inventory fluxes of the same location.

| Description | Year | Flux | Averaging area |
|---|---|---|---|
| Oktoberfest | 2018 | $6.7\,\mu g/(m^2 s)$ | $0.3\,km^2$ |
| TNO-MACC III | 2015 | $0.9\,\mu g/(m^2 s)$ | $4.6\,km^2$ |
| EDGAR v4.3.2 | 2012 | $1.0\,\mu g/(m^2 s)$ | $82\,km^2$ |
| IER | 2008 | $0.1\,\mu g/(m^2 s)$ | $4.0\,km^2$ |

The $CH_4$ share of the natural gas in Munich is on average about $96\,\%$ (München (2018c)). If we assume all of the $CH_4$ emissions are fossil-fuel related, the maximum loss rate can be determined as:

$$\frac{M_{CH4,loss,max}}{M_{CH4,total}} = \frac{1.46 \cdot 10^3\,kg}{1.34 \cdot 10^5\,kg \cdot 96\,\%} = 1.1\,\%. \tag{17}$$

This loss rate of $1.1\,\%$ is smaller than the gas leaks reported in the literature, such as a $2.7\,\%$ loss rate for the urban region of Boston (McKain et al. (2015)) or $2.3\,\%$ for the U.S. oil and gas supply chain (Alvarez et al. (2018)).

## 3.7 Comparison with existing $CH_4$ emission estimates

We are not aware of a comparable $CH_4$ study dealing with festivals. For a better illustration, we compared the determined emission flux of the Oktoberfest premises to the emission flux of Boston, which is known as a very leaky city. In the Boston study, McKain et al. (2015) quantified the regional averaged emission flux of $CH_4$ in Boston as $(18.5 \pm 3.7)\,g/(m^2 a)$ ($95\,\%$ confidence interval), which corresponds to $(0.6 \pm 0.1)\,\mu g/(m^2 s)$ and is less than a tenth of the emissions that we determined for the Oktoberfest premises. Although it is difficult to compare the small and densely populated Oktoberfest premises with the entire city area of Boston, the comparison shows that the emission flux of Oktoberfest is significant.

Furthermore, we compared the Oktoberfest emission flux to the state-of-the-art emission inventory fluxes of that particular area. For that purpose, the annual emission fluxes of TNO-MACC III (2015) (Denier van der Gon et al. (2017); Kuenen et al. (2014)), EDGAR v4.3.2 (2012) (Janssens-Maenhout et al. (2019)) and IER (2008) (Pregger et al.) are converted to the common unit $\mu g/(m^2 s)$. In Table 2, the converted values are shown. Furthermore, one can see that the different inventories have different spatial resolutions. Therefore, the fluxes are averaged over areas that represent not only the Oktoberfest premises but also additional urban districts. Nevertheless, it can be seen that the determined Oktoberfest emissions are significantly higher than all the three considered inventories. Therefore, festivals such as Oktoberfest can be significant $CH_4$ sources, although they are just present for a limited time of the year, and should be included in the inventories.



## 4 Conclusions and Outlook

In this study, the methane emissions at Oktoberfest 2018 in Munich were investigated. It is the first study that deals with the methane emissions of a big festival. We concentrated on Oktoberfest as it is the world's largest folk festival and a methane source that has not yet been taken into account in the state-of-the-art emission inventories.

Combining the in-situ measurements with a Gaussian plume dispersion model, the average emission number of Oktoberfest was determined to be $(6.7 \pm 0.6)\,\mu g/(m^2 s)$ ($1\sigma$ standard deviation). A comparison between weekdays $(4.6\,\mu g/(m^2 s))$ and weekend days $(8.5\,\mu g/(m^2 s))$ shows that the emission strength at the weekend was almost twice as high compared to during the week. It demonstrates that a higher number of visitors results in higher emissions. However, the daily emission cycle has an oscillating behavior that can not be explained by the number of visitors. These results suggest that $CH_4$ emissions at Oktoberfest not only come from the human biogenic emissions, which should be more than 10 times smaller according to our bottom-up calculation. It is more likely that fossil-fuel related emissions, such as incomplete combustion or loss in the gas appliances, are the major contributors to the Oktoberfest emissions.

Due to safety reasons, we were not allowed to enter the festival premises with the instrument. Therefore, the distance from the measurement points to the suspected sources on the festival terrain was large. This resulted in low $CH_4$ and $C_2H_6$ concentrations. Latter ones were even below the detection limit of the instrument. This limited the possibilities to attribute the emissions to specific sources. To improve this aspect, several additional approaches are possible for future studies. As we are not aware of a more sensitive portable $C_2H_6$ analyzer, discrete air sampling using sample bags within the tents for $C_2H_6$ and isotopic $CH_4$ measurements might be an option. Furthermore, the measurement of more tracers, such as CO, $CO_2$, etc., are additional possibilities to improve the source attribution. For other festivals it might also be a good option to go closer to the sources, which we were not permitted to do at Oktoberfest in Munich.

In summary, this study uses Oktoberfest as an exemplary event to show for the first time that big festivals can be significant $CH_4$ emitters. Therefore, these events should be included in future emission inventories. Furthermore, our results provide the foundation to develop reduction policies for such events and a new pathway to mitigate fossil fuel $CH_4$ emissions.

*Data availability.* All raw data can be provided by the authors upon request.

*Author contributions.* JC, FD and HM planned the campaign; JC, FD, HM, AF and DW performed the measurements; JC, FD, AF and DW analyzed the data; JC and FD wrote the manuscript draft; AF, MH, HDG and TR reviewed and edited the manuscript.

*Competing interests.* The authors declare that they have no conflict of interest.



*Acknowledgements.* We thank Peter Swinkels and his colleagues from Picarro Inc. who made it possible for us to borrow the GasScouter; Jörg Ochs (Munich municipal utilities, SWM) for providing us helpful information about the Munich gas distribution system; Hanns-Erik Endres, Manfred Engl and colleagues from Fraunhofer EMFT, Martin Christ (LMU) as well as the colleagues from Klenze Gymnasium Munich for providing access to measurement sites; Markus Garhammer, Matthias Wiegner and Mark Wenig (LMU) for providing us wind

5  and boundary layer height data; Anna-Leah Nickl and Mariano Mertens (DLR) for running high-resolution wind forecasts; Konrad Koch (TUM) for providing us detailed information about the Munich sewer system; Rachel Chang for fruitful discussions; our students Ankit Shekhar, Xiao Bi and Homa Ghasemifard for helping with the measurements; and Ankit Shekhar, Rita von Grafenstein for proofreading the paper.



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
