# Peer review of "Methane Emissions from the Munich Oktoberfest"

_Atmospheric Chemistry and Physics, 2019_

## Referee Comment (RC1) · Anonymous Referee #1 · 13 Nov 2019

General comments: This very original paper shows that methane emissions from large festivals such as the Munich Oktoberfest are measurable and non-negligible despite the time-limited event. In-situ methane measurements around the festival area are evaluated with plume modeling to assess the emissions and their uncertainties. The study about this somewhat amusing but also serious topic is comprehensive and robust. It is well written, concise, and contains informative figures. I therefore recommend publication after consideration of the following comments.

Specific comments:

p. 5 line 4: explain what is a Kaiser window, or give a reference

p. 5 line 7: explain why you chose 5 ppm as threshold, and how your results change when you choose another value.

[Figure]

Figure 8: by comparison with Fig 9 I would expect a third, certainly smaller yet distinct peak around 8.5 ug/(m2s) due to the weekend emissions. Why is this peak missing in Fig. 8? If Fig. 8 shows all emission estimates, then could it be that you did less measurements on a weekend day such that these measurements/samples are under-represented? In this case your overall emissions would be biased low. Maybe I have overlooked it: have you indicated how evenly in time your measurements were spaced, both over the course of the week, and over the course of the day (important for Fig. 11)? An additional figure could clarify this and eliminate doubts about a systematic bias due to possibly irregularly spaced measurement times. Also, in case of weekend under-representation, you could introduce weights to your measurements.

Figure 11: include the number of samples in the caption if it is constant, or as an additional line if not.

Technical corrections:

p. 5 line 6: better "minima", not "valleys"

p. 10 line 16: better "run", not "round"

---

## Referee Comment (RC2) · Anonymous Referee #2 · 26 Nov 2019

General comments This is a fascinating study of an amusing but very instructive topic. The methodology of the paper is simple yet very thorough and detailed. The work should be published.

I suspect the Oktoberfest does not rank in the world's top million methane sources (or would Oktoberfest indulgers say 'shourseshs'). So in itself, the problem is not especially important and applying very diligent and thoughtful analytical effort to a small problem is using a sledgehammer to crack a peanut.

But that snap impression misses the wider value of the work – this paper is a very thorough and careful exposition of how to quantify emissions from a somewhat-disseminated local clump of sources, using fairly simple techniques and relatively inexpensive instrumentation (and a bicycle!). The methodology developed here is applicable to a very wide range of similar sources, such as clumps of poorly regulated

tropical landfills, aggregations of cow barns, or variegated wetlands. Given that, this is potentially a very useful paper indeed.

Detailed comments Page 1 right L45 – also cite Etminan et al values? Page 2 right L84 – also give brief mention of human emissions, both eructated and flatulated? Page 3 left L12-13 – 'detection' – do you mean precision or detection threshold? Maybe rephrase this? Page 3 left L16 – I'm being pedantic but data re plural - 1 datum, 2 data – data 'are' not data 'is'. Page 3 right L59 – same pedant's comment – data were, not was. Page 4 left L19 – maybe expand this a little and rephrase? Is this a valid filtering method?- or could you be leaving out important burps of methane? Page 5 right L63 – define E here. Note – in line 74 just below, E has dimensions of g s-1 but later (e.g. P7 bottom line on right) the dimensions of E also include m2 – needs to be consistent. Page 9 right L40 – a huge range in the literature of human-produced methane. We've measured various communities and found great variety. Page 9 right L64. Not sure high flux means low emissions. Could also mean sudden filling of a lot of local side-ponds and water in cracks and holes, that don't flush and go anerobic, producing CH4. Page 10 left L27 – natural gas I assume? Not CO2 pushing the beer? Page 11 left L10-13 maybe rephrase a little – not too clear. And when you talk about 'bottom-up' emissions is this from humans? I thought the Oktoberfest was bottoms-down on seats? Page 11 left L28 One BIG omission that should be mentioned in the section of what to measure next is ISOTOPES. . .. . . ...

Page 11 left L37 – this paper has shown that comparatively simple methods can do a great deal to quantify emissions from clumps of methane sources, for example I'd think of groups of small cow barns, or uncovered heaps in badly managed tropical landfills, or wetlands made of groups of ponds and swamps. It's a really nice study investigating that type of problem and this should be brought out in the conclusion.

Overall – nice paper, fun to read, and potentially very widespread applications. The paper should be accepted and published with minor corrections.

---

## Author Comment (AC1) · 15 Jan 2020

Technische Universität München | Theresienstr. 90 | 80333 München

Munich, January 15, 2020

**Reponses to the comments of reviewer 1**

We would like to thank reviewer 1 for thoroughly reading our paper and providing very helpful and insightful comments. Below, please find our responses to the reviewer's comments.

**General comments**:

This very original paper shows that methane emissions from large festivals such as the Munich Oktoberfest are measurable and non-negligible despite the time-limited event. In-situ methane measurements around the festival area are evaluated with plume modeling to assess the emissions and their uncertainties. The study about this somewhat amusing but also serious topic is comprehensive and robust. It is well written, concise, and contains informative figures. I therefore recommend publication after consideration of the following comments.

**Response**: *Thank you very much for appreciating our work and supporting its publication.*

**Specific comments:**

1. p. 5 line 4: explain what is a Kaiser window, or give a reference

**Response:** *We have included a reference to explain the Kaiser window:*

*Kaiser, James F.; Schafer, Ronald W. (1980). "On the use of the $I_0$-sinh window for spectrum analysis". IEEE Transactions on Acoustics, Speech, and Signal Processing.* ***28****: 105–107.* *[doi](https://...):[10.1109/TASSP.1980.1163349](https://...).*

2. p. 5 line 7: explain why you chose 5 ppb as threshold, and how your results change when you choose another value

**Response:** *We introduced a new sentence in the paper to explain why we chose the 5 ppb: "We chose this threshold to be equal to the combined uncertainty of the instrument (3 ppb) and background (4 ppb)"*

3. Figure 8: by comparison with Fig 9 I would expect a third, certainly smaller yet distinct peak around 8.5 ug/(m2s) due to the weekend emissions. Why is this peak missing in

**Technische Universität München**
Fakultät für Elektrotechnik
Professur für Umweltsensorik und
Modellierung

**Prof. Dr.-Ing.**
**Jia Chen**
Theresienstr. 90
80333 München

jia.chen@tum.de
www.esm.ei.tum.de
www.tum.de

Tel. +49 89 289-23350

Bayerische Landesbank
IBAN-Nr.:
DE10700500000000024866
BIC: BYLADEMM
Steuer-Nr.: 143/241/80037
USt-IdNr.: DE811193231

Fig. 8? If Fig. 8 shows all emission estimates, then could it be that you did less measurements on a weekend day such that these measurements/samples are under-represented? In this case your overall emissions would be biased low.

**Response:** *We appreciate your thoughts. However, in Fig. 8 we show the distribution of the averaged emission numbers over the whole Oktoberfest time period, including during the week and the weekend. The spread of the Gaussian curve results from the uncertainty of the input signals, i.e. wind, background, measurement error. We have changed the first sentence of section 3.2, to make it clearer. The question about the bias is answered in the next comment.*

4. Maybe I have overlooked it: have you indicated how evenly in time your measurements were spaced, both over the course of the week, and over the course of the day (important for Fig.11)? An additional figure could clarify this and eliminate doubts about a systematic bias due to possibly irregularly spaced measurement times. Also, in case of weekend under- representation, you could introduce weights to your measurements.

**Response:** *The daily distribution of the measurements has been included to Figure 11. Furthermore, we included the following sentence to make clear how many plumes were recorded on weekdays and weekends, respectively: "After grouping the emission numbers into the two categories, weekday (in total 59 valid plumes) and weekend (26 valid plumes), two separated distributions are visible in Figure 9."*
*The ratio between measurements at weekend days and weekdays is about 2 to 5, which is similar to their respective occurrence in a 7-day week. Therefore, we don't think that weekend days are underrepresented and need additional weighting factors.*

5. Figure 11: include the number of samples in the caption if it is constant, or as an additional line if not.

**Response:** *Figure 11 was changed. It contains now the number of valid plumes for each hour.*

6. p. 5 line 6: better "minima", not "valleys"

**Response:** *We changed it to "minima" according to your suggestion*

7. p. 10 line 16: better "run", not "round"

**Response:** *We changed it to "run" according to your suggestion*

With best regards,

Jia Chen, Florian Dietrich on behalf of all co-authors

---

## Author Comment (AC2) · 15 Jan 2020

Munich, January 15, 2020

**Answers to the comments of reviewer 2**

We would like to thank reviewer 2 for thoroughly reading our paper and providing very helpful and insightful comments. Below, please find our responses to the reviewer's comments.

**General comments:** This is a fascinating study of an amusing but very instructive topic. The methodology of the paper is simple yet very thorough and detailed. The work should be published. I suspect the Oktoberfest does not rank in the world's top million methane sources (or would Oktoberfest indulgers say 'shourseshs'). So in itself, the problem is not especially important and applying very diligent and thoughtful analytical effort to a small problem is using a sledgehammer to crack a peanut. But that snap impression misses the wider value of the work – this paper is a very thorough and careful exposition of how to quantify emissions from a somewhat disseminated local clump of sources, using fairly simple techniques and relatively inexpensive instrumentation (and a bicycle!). The methodology developed here is applicable to a very wide range of similar sources, such as clumps of poorly regulated landfills, aggregations of cow barns, or variegated wetlands. Given that, this is potentially a very useful paper indeed.

**Response:** *We are very grateful for reviewer's recognition of our paper's novelty and potential impact. Thank you very much for supporting the publication of this work*.

**Detailed comments:**

1. Page 1 right L45: also cite Etminan et al values?

**Response:** *We have added GWP is even 14% higher according to Etminan et al. (2016).*

2. Page 2 right L84: – also give brief mention of human emissions, both eructated and flatulated?

**Response:** *Thank you for this suggestion. In the paragraph that you have mentioned, we have listed official statistics about Oktoberfest. However, there are no statistics/measurements about human emissions in Oktoberfest. We have performed calculations of human emissions based on literature values, which is discussed in section 3.4.*

**Technische Universität München**
Fakultät für Elektrotechnik
Professur für Umweltsensorik und
Modellierung

**Prof. Dr.-Ing.**
**Jia Chen**
Theresienstr. 90
80333 München

Tel. +49 89 289-23350

jia.chen@tum.de
www.esm.ei.tum.de
www.tum.de

Bayerische Landesbank
IBAN-Nr.:
DE10700500000000024866
BIC: BYLADEMM
Steuer-Nr.: 143/241/80037
USt-IdNr.: DE811193231

3. Page 3 left L12-13: – 'detection' – do you mean precision or detection threshold? Maybe rephrase this?

**Response:** *we rephrased the sentence to "using a laser as a light source and a high-finesse optical cavity for measuring gas concentrations with high precision, which is 3 ppb for the CH4 mode with 1 s integration time. "*

4. Page 3 left L16: Page 3 left L16 – I'm being pedantic but data re plural - 1 datum, 2 data– data 'are' not data 'is'.

**Response:** W*e have changed the sentence to "data are averaged"*

5. Page 3 right L59: same pedant's comment – data were, not was.

**Response:** *We rewrote the sentence completely so that there is no "data" in the sentence anymore.*

6. Page 4 left L19: maybe expand this a little and rephrase? Is this a valid filtering method? - or could you be leaving out important burps of methane?

**Response:** *You are right; we are losing some of the rounds by applying this filtering method. However, those rounds are in our opinion no valid rounds and, therefore, are negligible. In our approach, the shape of the plume determined by the forward model (for exactly one wind direction) is compared to the actual shape of the measured plume. In case the wind is very variable, it does physically not make sense anymore to compare the modeled and measured plume shape with each other anymore. Therefore, we optimally need calm wind conditions. As this is practically not possible and as we do not know the wind direction exactly (24° standard deviation), we decided to use this value as a hard threshold to filter out plumes. To make it clear, we added the following sentence in the paper: "Those 24° represent the measurement uncertainty in the wind direction (cf. 2.2.6) and are therefore well suited as a lower limit for filtering out too variable wind conditions."*

7. Page 5 right L63: – define E here. Note – in line 74 just below, E has dimensions of g s-1 but later (e.g.P7 bottom line on right) the dimensions of E also include m2 – needs to be consistent.

**Response:** *Thank you very much for pointing this out.*
*We introduced $E_{Okt,i}$ in the sentence as "[…] considered for the determination of the emission number of Oktoberfest $E_{Okt,i}$". The units of E have been unified as well. We modified line 74 to "3 µgs-1m-2".*

8. Page 9 right L40: a huge range in the literature of human-produced methane. We've measured various communities and found great variety.

**Response:** *Thank you for the hint. We have mentioned the previous literature for human $CH_4$ emission via breath and flatus surveyed in Polag et al. 2019 in the paper. We have also*

*modified our calculation for human biogenic emissions according to the values reported in Polag et al. 2019 (Please see section 3.4).*

9. Page 9 right L64: Not sure high flux means low emissions. Could also mean sudden filling of a lot of local side-ponds and water in cracks and holes, that don't flush and go anaerobic, producing CH4.

**Response:** *Thank you for pointing out this possibility. Nevertheless, we do not think that filling of local side-ponds is the most likely event that will happen if there is a high wastewater flux. The Munich sewer system is a gravity sewer and, therefore, most of the water should flow all the time based on gravity. Therefore, the effect of methane production in case of high wastewater fluxes is in our opinion not dominating in such a sewer system.*

10. Page 10 left L27: natural gas I assume. Not CO2 pushing the beer?

**Response:** *Thank you. We changed it to "natural gas".*

11. Page 11 left L10-13: maybe rephrase a little – not too clear. And when you talk about 'bottom-up' emissions is this from humans? I thought the Oktoberfest was bottoms-down on seats?

**Response:** *Thank you for this note! We change the sentence to "These results suggest that $CH_4$ emissions at Oktoberfest do not come solely from the human biogenic emissions, which was according to our calculation 5 times smaller than the emissions determined for the Oktoberfest."*

12. Page 11 left L28: One BIG omission that should be mentioned in the section of what to measure next is ISOTOPES...

**Response:** *Thank you. We changed the sentence to "Furthermore, the measurement of isotope ratios, such as $\delta 13C$ and $\delta D$ are useful options to improve the source attribution."*

13. Page 11 left L37: this paper has shown that comparatively simple methods can do a great deal to quantify emissions from clumps of methane sources, for example I'd think of groups of small cow barns, or uncovered heaps in badly managed tropical landfills, or wetlands made of groups of ponds and swamps. It's a really nice study investigating that type of problem and this should be brought out in the conclusion.

**Response:** *We really appreciate you raising this point. Thank you very much for your approval of our method. We have added a paragraph in the conclusion emphasizing the wide applicability of this method.*

With best regards,

Jia Chen, Florian Dietrich on behalf of all co-authors